# Effects of Dietary Supplementation with *Abies sibirica* Essential Oil on Growth Performance, Digestive Enzymes, Skin Mucus Immunological Parameters, and Response to Heat Stress in Rainbow Trout

**DOI:** 10.3390/ani15192911

**Published:** 2025-10-07

**Authors:** Morteza Yousefi, Hossein Adineh, Yury Anatolyevich Vatnikov, Evgeny Vladimirovich Kulikov, Olesya Anatolyevna Petrukhina, Elena Dmitriyevna Sotnikova, Alena Igorevna Telezhenkova, Seyyed Morteza Hoseini

**Affiliations:** 1Department of Veterinary Medicine, RUDN University, 6 Miklukho-Maklaya St., 117198 Moscow, Russia; vatnikov-yua@rudn.ru (Y.A.V.); kulikov-ev@rudn.ru (E.V.K.); petrukhina-oa@rudn.ru (O.A.P.); sotnikova-ed@rudn.ru (E.D.S.); telezhenkova-ai@rudn.ru (A.I.T.); seyyedmorteza.hoseini@gmail.com (S.M.H.); 2Department of Fisheries, Faculty of Agriculture and Natural Resources, Gonbad Kavous University, Gonbad Kavous 4971799151, Golestan, Iran; adineh.h@gmail.com; 3Inland Waters Aquatics Resources Research Center, Iranian Fisheries Sciences Research Institute, Agricultural Research, Education and Extension Organization, Gorgan 4916687631, Golestan, Iran

**Keywords:** thermal shock, phytogenic, Siberian fir, mucosal immunity, oxidative stress

## Abstract

**Simple Summary:**

This study addresses the impact of dietary essential oil from the Siberian fir on growth performance, overall health, and heat stress resistance in rainbow trout. The essential oil was added at 100, 200, and 400 mg/kg to diets for 60 days. The fish that received the essential oil showed improved growth, higher gut digestive enzymes, stronger immune and antioxidant responses, and improved resistance to heat stress compared to the those received an unsupplemented diet. The best results were obtained in fish given essential oil at the 100 mg/kg level. This treatment reduced physiological stress responses and oxidative stress in the fish after the heat stress. Overall, adding Siberian fir essential oil to fish diets at the 100 mg/kg level can help rainbow trout grow better and be more resilient to heat stress.

**Abstract:**

Climate change and global warming are concerning issues impacting various industries. In the aquaculture industry, these issues are more important in coldwater species, like rainbow trout *Oncorhynchus mykiss*. Hence, strategies to control these negative effects are worthy of study. Herbal feed additives are reliable tools to increase fish growth and health, thereby mitigating the drawbacks of climate change on fish. In this study, three diets containing 100 (100EO), 200 (200EO), and 400 (400EO) mg/kg essential oil of *Abies sibirica* (SBF) along with a control diet (CTL; unsupplemented) were fed to triplicate groups of fish for 60 days. Then the fish were exposed to a 96 h heat stress (25 °C) to monitor their survival and biochemical responses. The results showed that growth performance, feed efficiency, heat stress resistance, intestinal activity of digestive enzymes, and skin mucus immunological parameters significantly (*p* < 0.05) increased in the SBF essential oil treatments, and the highest increases were observed in the 100EO treatment, followed by the 200EO group. Dietary supplementation with SBF essential oil significantly (*p* < 0.05) mitigated heat stress-induced increases in plasma cortisol and glucose. Moreover, dietary SBF essential oil significantly (*p* < 0.05) enhanced immunological parameters such as plasma and intestinal lysozyme and immunoglobulin levels, and improved hepatic antioxidant defenses (including catalase, glutathione peroxidase, total antioxidant capacity, and reduced glutathione), while reducing lipid peroxidation. These effects were most pronounced in the 100EO and 200EO treatments, with the highest performance being observed in the former group. In conclusion, dietary SBF essential oil at 100 mg/kg is capable of augmenting growth performance, immunity, and antioxidant capacity, and suppressing physiological stress, thereby augmenting fish resilience against heat stress.

## 1. Introduction

Rainbow trout, *Oncorhynchus mykiss*, is an important coldwater fish species with annual production being around 900,000 tons throughout the world [1]. Although it is a relatively hardy fish species, high water temperatures are a serious threat to it [2]. Climate change, particularly the increase in global temperature, is one of the critical challenges confronting the global aquaculture industry [3]. Ectothermic organisms, such as fish, are directly influenced by water temperature. An elevation in water temperature results in a decrease in water dissolved oxygen levels [4] and concurrently leads to an increase in the basal metabolic rate of aquatic animals [5]. These conditions not only heighten energy consumption but also promote the production of reactive oxygen species during cellular respiration, ultimately resulting in diminished growth and health in fish [2,3]. Moreover, the occurrence of heatwaves in various regions worldwide exerts short-term yet acute effects on aquaculture. Such heatwaves can induce rapid increases in water temperature, leading to acute heat stress in fish, characterized by increases in blood cortisol and glucose, oxidative stress, and immunosuppression, ultimately leading to an increase in fish susceptibility to diseases [2,6,7,8].

Consequently, it is imperative to identify and implement strategies to mitigate the impacts of climate change and rising temperatures on aquaculture. A promising approach to enhance the growth and resilience of aquatic organisms against various stressors is the use of dietary additives [9]. Among these additives, herbal supplements are particularly noteworthy due to their high antioxidant properties and their ability to stimulate immune responses in aquatic animals [10]. Furthermore, these supplements have demonstrated significant positive effects on disease resistance, reduction in stress-related impacts on fish, and the enhancement of growth performance [11,12]. For example, dietary supplementation with essential oils from various plants, such as oregano [13] and *Artemisia dracunculus* [14], could improve the fish growth performance. Moreover, dietary supplementation with thymol [2] and *Hyssopus officinalis* extract [15] could increase heat stress resistance in rainbow trout, along with improvements in non-specific immunity and antioxidant capacity. However, there are many herbal additives not studied in fish nutrition yet, which gives rise to the need for further studies in this field.

Siberian fir (SBF), *Abies sibirica*, is a coniferous evergreen tree found frequently in the northern Russia, where it forms the main structure of the dark coniferous taiga [16]. The essential oil of SBF has exhibited various biological activities, including radical-scavenging and antimicrobial potentials [16,17], which make it a suitable feed additive candidate. Despite these potentials, the effects of SBF essential oil have not been studied in aquatic animals, yet. We hypothesized that SBF essential oil can improve heat resistance in rainbow trout by suppressing oxidative stress. Moreover, we hypothesized that this essential oil can improve the growth rate and innate immunity of the fish, as previously observed in other studies on various essential oils. Thus, this study aimed to assess the effects of dietary SBF essential oil on growth performance, antioxidant and immunological responses, and resistance to heat stress in rainbow trout.

## 2. Materials and Methods

### 2.1. SBF Essential Oil Origin and Composition

SBF essential oil was obtained from Mirrolla LLC., located in Moscow, Russia, and its composition was determined using Gas Chromatography–Mass Spectrometry (GC-MS). It was found that 14-β-H-Pregna (39.68%), 2-Octadecoxyethanol (12.36%), α-pinene (10.36%), and eicosane (1.05%) were the main components of the essential oil (Table 1).

### 2.2. Diet Formulation and Feeding Trial

SBF essential oil was incorporated into the diet at three different levels: 100 mg/kg (100EO), 200 mg/kg (200EO), and 400 mg/kg (400EO). A diet without SBF essential oil supplementation also served as a control (CTL). To formulate the diets, the feed ingredients listed in Table 2 were powdered and sieved (200 μ), then mixed in an electric mill (Model UM-120; Unique Co., Zhengzhou, China) for 20 min. SBF essential oil was mixed with dietary oils (sunflower/soybean oil) before being added to the feed ingredient mixture. All ingredients were mixed with 30% water to create a uniform paste. The resulting paste was pelletized (2 mm diameter) using a meat grinder (Model: No. 32; Fajr Co., Theran, Iran) and dried by a fan blower. The proximate composition of the diets was determined using standard methods [18] prior to the initiation of the feeding trial, and the produced pellets were stored at 4 °C until use.

Experimental fish were commercially procured and maintained in university’s aquaculture research facility. Initially, the fish were acclimatized to the laboratory conditions for 10 days, during which they were fed the CTL diet. Subsequently, 240 fish were stocked in 12 tanks of 60 L each, at a density of 20 fish per tank. The aforementioned diets were offered to the fish over 60 days, with a daily feeding rate of 3% of biomass, calculated and readjusted biweekly. All tanks were continuously aerated throughout the experiment, and fish waste was daily siphoned. During the siphoning, 30% of the tank water was replaced by clean water (dechlorinated tap water). Water quality parameters, including temperature (17.0 ± 0.86 °C), dissolved oxygen (8.51 ± 0.33 mg/L), pH (7.70 ± 0.11), and total ammonia (0.54 ± 0.03 mg/L), were measured during the experiment.

At the end of the feeding trial, fish growth performance and survival, and feed efficacy were calculated in each treatment based on the following formulas:$$Specific\;growth\;rate\;\left(\%/d\right)\;=\;100\;\times \;\frac{ln\;(final\;weight)\;-\;ln\;(initial\;weight)}{rearing\;period\;(days)}$$$$Weight\;gain\;\left(\%\right)\;=\;100\;\times \;\frac{final\;weight\;-\;initial\;weight}{initial\;weight}$$$$Feed\;conversion\;ratio\;=\;\frac{feed\;intake}{final\;weight\;-\;initial\;weight}$$$$Survival\;(\%)\;=\;100\;\times \;\frac{final\;fish\;count\;in\;tank}{Initial\;fish\;count\;in\;tank}$$

### 2.3. Heat Stress

Upon the termination of the feeding trial, ten fish from each aquarium were subjected to a short-term heat stress. The water temperature was gradually elevated from 17 °C to 25 °C over a 12 h period (at a rate of 1 °C every 1.5 h) using submersible heaters (DW-FY-50W heater; Zhongshan Songbao Electric Appliance Co., Ltd.; Lianfeng Road Xiaolan Town, China). A previous study has shown that this protocol of heat stress causes near 50% mortality and changes to the physiological state of rainbow trout [2]. The fish were maintained at 25 °C for 96 h to evaluate their heat resistance, during which dead fish were removed daily. To assess the physiological responses to heat stress, samples of liver, blood, and intestine were collected after 24 h of exposure to the elevated temperature, as detailed in the following sections.

### 2.4. Sampling

The intestinal activity of digestive enzymes and skin mucus immunological parameters were monitored exclusively at the end of the feeding trial (before the heat stress). Meanwhile, plasma stress and immunological parameters, intestinal immunological parameters and hepatic antioxidant parameters were analyzed before and after the heat stress. At each sampling time (i.e., before and after the heat stress), three fish were caught per tank and anesthetized in a 100 ppm eugenol bath. Fish skin mucus was collected indirectly by placing a fish in a polyethylene bag containing 5 mL of NaCl solution (0.85%). After 60 s and gentle rubbing, the fish was removed from the bag and a mucus sample was collected in a plastic tube. The mucus extract was obtained by centrifugation (15 min at 13,000 rpm and 4 °C) and kept at −70 °C until analysis. The fish blood was taken from the caudal vein using heparinized syringes and centrifuged (10 min at 5000 rpm and 4 °C) for plasma separation. The plasma specimens were kept at −70 °C until analysis. After blood sampling, the fish were killed by a sharp blow on the head and spinal cord severance. The abdominal cavity of the fish was opened and pieces of anterior gut (for a digestive enzyme assay), posterior gut (for an immunological assay), and liver (for an antioxidant assay) were dissected and frozen in liquid nitrogen. The collected tissues were homogenized in one volume of phosphate buffer (pH 7.0) and centrifuged (15 min at 13,000 rpm and 4 °C) to obtain enzyme extracts. The enzyme extracts were kept at −70 °C until analysis. Subsamples (obtained from three fish per tank) were pooled to generate three replications per treatment.

### 2.5. Analysis

#### 2.5.1. Digestive Enzymes

The gut amylase activity was assessed in the enzyme extract through the hydrolysis of alpha starch, following the methodology outlined previously [19]. The activity of gut lipase was evaluated by monitoring the degradation of 1,2-O-dilauryl-rac-glycero-3-glutaric acid (6-methyl-resorophine), as described by Iijima et al. [20]. Furthermore, gut protease activity was quantified using AZOcasein as a substrate, in accordance with the methods established by Iversen and Jørgensen [21]. The activities of these enzymes were expressed as specific activity, which is defined as the enzyme activity per unit of soluble protein concentration in the enzyme extract. The soluble protein concentration was determined by Lowry et al.’s [22] method.

#### 2.5.2. Skim Mucus and Intestinal Immune-Related Parameters

The skin mucus peroxidase activity was assessed using 3,3′,5,5′-tetramethylbenzidine hydrochloride in conjunction with hydrogen peroxide as substrates, with measurements taken at a wavelength of 450 nm, as described by Quade and Roth [23]. For the evaluation of the mucus protease activity, AZOcasein served as the substrate, and the rate of hydrolysis was quantified at 350 nm, following the protocol established by Iversen and Jørgensen [21].

Skin mucus and intestinal lysozyme activities were measured as described by Ellis [24]. The mucus extract was mixed with a suspension of *Micrococcus luteus* (in phosphate buffer pH 6.2) and the decrease in optical density was recorded for 5 min at 500 nm. Each 0.001 decrease in the optical density per minute was considered to be one unit of lysozyme activity. Additionally, skin mucus alkaline phosphatase (ALP) activity was determined using a commercial kit provided by Man Co. (Tehran, Iran), following the procedures described by Esmaeili et al. [25]. The activities of these enzymes were expressed as specific activity, which is defined as the enzyme activity per unit of soluble protein concentration in the enzyme extract. The soluble protein concentration was determined using the Lowry method, as described by Lowry, Rosebrough, Farr, and Randall [22].

The skin mucus and intestinal total immunoglobulin (Ig) levels were determined as described by Siwicki and Anderson [26]. The mucus extract was mixed with polyethylene glycol solution (12%) and shaken for 2 h to precipitate Ig. Then, the mixture was centrifuged and the soluble protein concentration was measured in the supernatant according to Lowry, Rosebrough, Farr, and Randall [22]. The difference in the soluble protein levels of the original extract and the precipitated one was equal to the total Ig level.

#### 2.5.3. Plasma Immune-Related Parameters

Plasma lysozyme activity was measured as stated above for the skin mucus sample, but expressed as units per volume of the sample. Plasma total Ig levels were measured as described above for the skin mucus sample. Plasma alternative complement (ACH50) activity was determined by measuring the hemolytic activity of the samples against sheep erythrocyte. The reaction mixture contained EGTA, magnesium, and gelatine in a barbital buffer (pH 7.0). Diluted plasma (0.306–10% in the same buffer) was mixed with the sheep erythrocyte and incubated at room temperature for 90 min. The hemolysis rate was measured by recording the optical density of the supernatant at 412 nm, and ACH50 activity was measured according to Yano [27].

#### 2.5.4. Hepatic Antioxidant-Related Parameters

Antioxidant parameters were evaluated using commercial kits provided by Zellbio Co. (Deutschland, Germany). The activity of superoxide dismutase (SOD) was quantified based on the autoxidation of pyrogallol, as recommended by Marklund [28]. Catalase (CAT) activity was assessed by measuring the rate of hydrogen peroxide decomposition, following the methodology described by Goth [29]. The activities of glutathione peroxidase (GPx) and the levels of reduced glutathione (GSH) were determined through reactions with Ellman’s reagent, as previously described by Hu [30] and Sattar et al. [31]. Total antioxidant capacity (TAC) was measured based on the sample’s ability to reduce ferric ions, as detailed by Lim and Lim [32]. The concentration of malondialdehyde (MDA) was assessed by measuring the MDA-thiobarbiturate adduct at 95 °C, following the method suggested by Buege and Aust [33]. The activities of the antioxidant enzymes were expressed as specific activities, as described above for skin mucus parameters.

### 2.6. Statistical Analysis

Data related to fish growth performance, digestive enzyme activities, skin mucus immunological parameters, and post-stress survival were analyzed using one-way analysis of variance (ANOVA), followed by Duncan’s multiple range test. In contrast, plasma and gut biochemical and immunological parameters, along with hepatic antioxidant parameters, were analyzed using repeated measures two-way ANOVA (diet × stress) in conjunction with Duncan’s tests. Prior to conducting the ANOVA analyses, the normal distribution of the data and the homogeneity of variances among treatments were verified using the Shapiro–Wilk and Levene tests, respectively. Additionally, post-stress survival data were arcsin-transformed prior to analysis. The significance level was set at *p* < 0.050, and all statistical analyses were performed using SPSS version 21.

## 3. Results

Dietary supplementation with SBF essential oil significantly enhanced the growth performance of the fish. The final weight of the fish increased significantly across all SBF treatments, with significant improvements in weight gain, specific growth rate, and feed conversion ratio being observed in the 100EO and 200EO treatments. The highest final weight, weight gain, and specific growth rate were recorded in the 100EO treatment, while the best feed conversion ratios were noted in both the 100EO and 200EO treatments. No mortalities were found among the treatments (Table 3).

The dietary treatments also had a significant impact on the activity of gut digestive enzymes. The activities of amylase, protease, and lipase in the guts of the fish were significantly elevated in the 100EO and 200EO treatments, although no such increase was observed in the 400EO treatment. The highest enzyme activities were recorded in the 100EO treatment (Figure 1).

Furthermore, the dietary treatments significantly affected the skin mucus immunological parameters. Both lysozyme and peroxidase activities in the skin mucus were significantly higher in the 100EO and 200EO treatments compared to the CTL group, with the highest lysozyme activity observed in the 100EO treatment. Additionally, skin mucus ALP and protease activities, as well as total Ig levels, significantly increased across all SBF essential oil treatments. The highest ALP activities were noted in the 100EO and 200EO treatments, while the 100EO treatment exhibited the highest protease activity and total Ig level (Figure 2).

Dietary SBF essential oil significantly affected fish survival after the heat stress. The 100EO and 200EO treatments demonstrated significantly higher post-stress survival rates compared to the CTL group, with the highest survival observed in the 100EO treatment (Figure 3).

Dietary supplementation with SBF essential oil and heat stress significantly affected plasma cortisol levels. While heat stress resulted in a significant increase in plasma cortisol levels, dietary SBF essential oil mitigated such increases. There was an interaction effect of SBF essential oil and heat stress on plasma glucose levels. Plasma glucose levels were comparable across the 100EO, 200EO, and 400EO treatments before the stress. Plasma glucose levels significantly increased in all treatments after the heat stress; however, the 100EO treatment exhibited a significantly lower plasma glucose level compared to the CTL and 400EO treatments. Moreover, the plasma glucose levels in the 200EO treatment were significantly lower than in the CTL group after the stress (Table 4).

There were interactions between dietary SBF essential oil and heat stress on plasma and intestinal immunological parameters. Plasma lysozyme and ACH50 activities, as well as total Ig levels, significantly increased in the SBF essential oil treatments both before and after the stress. Conversely, the stress significantly reduced plasma ACH50 activity and total Ig levels across all treatments. The highest plasma lysozyme activity and total Ig level were recorded in the 100EO treatment, both before and after the stress. The highest plasma ACH50 activity was also observed in the 100EO treatment before heat stress, while the highest activity after stress was noted in both 100EO and 200EO treatments (Table 5).

Intestinal lysozyme activity significantly increased in the SBF essential oil treatments both before and after heat stress. The 200EO treatment exhibited the highest intestinal lysozyme activity prior to heat stress. However, heat stress significantly decreased intestinal lysozyme activities in the CTL, 200EO, and 400EO treatments. After the stress, the 100EO treatment demonstrated the highest intestinal lysozyme activity. Additionally, intestinal total Ig levels significantly increased in the 100EO and 200EO treatments both before and after stress exposure. Nevertheless, heat stress significantly reduced intestinal total Ig levels across all treatments, with the 200EO and 100EO treatments showing the highest total Ig levels, respectively, before and after stress (Table 5).

Before the stress, the 200EO and 400EO treatments exhibited significantly lower hepatic SOD activities compared to the CTL group. Following the stress, hepatic SOD activities were significantly reduced in the CTL, 100EO, and 200EO treatments, with the highest post-stress hepatic SOD activity recorded in the 100EO treatment.

Hepatic CAT activity significantly increased in the 200EO and 400EO treatments before the stress. However, the stress resulted in a significant increase in hepatic CAT activities in the CTL and 100EO treatments, while a decrease was observed in the 200EO and 400EO treatments. The SBF essential oil treatments demonstrated significantly higher hepatic GPx activities both before and after stress exposure. The heat stress significantly decreased hepatic GPx activity in the CTL, 100EO, and 400EO treatments, while an increase was observed in the 200EO treatment. The highest hepatic GPx activities were recorded in the 100EO and 200EO treatments before and after stress, respectively (Figure 4).

Additionally, the SBF essential oil treatments exhibited significantly higher hepatic TAC and GSH levels both before and after stress exposure. Heat stress significantly decreased hepatic TAC in the CTL group, while it increased hepatic GSH levels in the 100EO and 200EO treatments. The highest hepatic TAC and GSH levels were observed in the 100EO and 200EO treatments, respectively, both before and after stress.

SBF essential oil significantly lowered hepatic MDA levels prior to and after stress exposure. The stress significantly elevated hepatic MDA levels in the CTL, 200EO, and 400EO treatments. Before stress, the lowest hepatic MDA level was recorded in the 200EO treatment, whereas after stress, the lowest hepatic MDA levels were observed in the 100EO and 200EO treatments (Figure 4).

## 4. Discussion

Dietary phytogenic additives are recognized for their growth-promoting effects in aquaculture. These additives enhance fish growth performance through various mechanisms, including improved feed digestion, which increases the availability of nutrients for absorption and utilization [34]. Consequently, the elevated activity of digestive enzymes can be an indicator of enhanced digestion efficiency. Additionally, a reduced basal metabolic rate allows for greater energy allocation towards growth [35]. Fish in aquaculture farms often experience various stressors such as confinement, overcrowding, and deteriorating water quality [36]. Therefore, stress mitigation through dietary administration of phytogenic additives may have contributed to an improvement in growth performance.

The present study demonstrated that dietary supplementation with SBF essential oil significantly increased digestive enzyme activity while simultaneously decreasing plasma cortisol and glucose levels. These results suggest that the observed enhancement in growth performance may be attributed to improved nutrient digestion and reduced stress and energy expenditure. These findings align with previous research on rainbow trout, which has indicated that diets supplemented with essential oils from various plants, such as oregano [13] and *A. dracunculus* [14], improved fish growth performance.

Diseases pose a significant threat to the aquaculture industry and a robust surface mucosal immune defense is crucial for reducing the likelihood of disease outbreaks in fish farms [37]. Water serves as the primary route for disease transmission, so the skin mucus immune system plays a vital role in preventing pathogen entry into the fish body. Herbal supplements have been shown to enhance skin mucus immunity [38,39,40], as observed in the present study. Basal Ig levels in skin mucus are essential for pathogen detection and elimination [41]. Increases in skin mucus lysozyme and peroxidase activities enhance the germicidal capacity of the mucus [42,43]. Furthermore, elevated protease activity reduces pathogen adherence and colonization on the mucus surface [43], while ALP activity helps mitigate inflammation caused by potential pathogens [44]. Collectively, these changes may increase the skin power to eliminate pathogens and resistance against their negative effects. However, further studies with pathogenic challenge can address precisely this.

Cortisol, often referred to as the stress hormone in fish, leads to heightened energy expenditure, which is characterized by hyperglycemia, and results in immunosuppression [45]. Stress is prevalent in aquaculture facilities and activates various physiological responses, including increased basal energy expenditure, catabolism, cell respiration, and immunosuppression [45,46]. Dietary herbal additives are recognized for their antistress properties, which promote growth rate and immunocompetence [36]. The present study showed that SBF essential oil decreases plasma cortisol and glucose levels, indicating antistress property of the essential oil, which may be a reason for higher growth rate and immunological parameters in the fish. Similarly, previous studies have shown that other essential oils suppressed plasma cortisol and glucose levels in rainbow trout [14,47].

Innate immunity plays a vital role in disease prevention in fish [48], and the stimulation of this immune system through dietary additives has been recognized as an effective strategy for disease prevention [49]. Components of the innate immune system are distributed across various tissues, including blood and the intestine [48]. Humoral innate immune components function as systemic mechanisms to combat pathogens [50]. Additionally, the fish gut serves as a significant immunological tissue due to its direct exposure to the surrounding water and its unique mucosal immune characteristics [51]. Research has demonstrated that phytogenic feed additives can enhance both systemic and intestinal immunity in rainbow trout [52,53], thereby supporting the immunomodulatory effects of SBF essential oil observed in the present study for the first time. However, whether this activated innate immunity can boost fish disease resistance or lead to immune fatigue needs further studies including pathogenic challenges.

Dietary herbal supplements are rich in natural reducing and radical scavenging compounds that directly neutralize free radicals, thereby exerting antioxidant effects in fish [54]. Numerous studies have demonstrated that these additives stimulate the activity of antioxidant enzymes, alleviating oxidative stress in fish [55,56,57]. The present study identified 14-β-H-Pregna, α-pinene, and eicosane as the primary components of SBF essential oil. Both α-pinene and eicosane have been shown to possess radical-scavenging and antioxidant properties [58,59,60]. Additionally, essential oils rich in 14-β-H-Pregna have demonstrated similar effects [61]. Therefore, SBF essential oil is capable of scavenging free radicals while simultaneously enhancing antioxidant enzyme activity, contributing to increased antioxidant reserves and reduced lipid peroxidation.

As a coldwater fish species, rainbow trout is particularly sensitive to increases in water temperature, prompting research into the use of phytogenic additives to enhance heat resistance [2]. The present study demonstrates for the first time that SBF essential oil can improve heat stress resilience in rainbow trout. Previous studies have reported similar outcomes with other phytogenic feed additives, such as thymol [2] and *H. officinalis* extract [15]. Thermal stress is linked to physiological and oxidative stress, resulting from elevated metabolic rate and increased formation of reactive oxygen species [3]. The present findings indicate that the enhanced heat stress resistance observed in the SBF essential oil treatments is associated with the antioxidant and antistress properties of the essential oil, as discussed above, and is partially corroborated by earlier research [2,15]. Furthermore, SBF essential oil was effective in mitigating immunosuppression following heat stress, thereby potentially improving the resistance to diseases of the fish. However, it should be noted that fish recovery after heat stress was not addressed in the present study and it is worth evaluating physiological recovery (or potential delayed mortalities) during recovery in future studies, as reported in other studies [62,63].

## 5. Conclusions

In conclusion, SBF essential oil can be used as a feed additive in rainbow trout aquaculture. The essential oil promotes the growth performance of fish, which may be due to elevations in the activity of digestive enzymes and mitigated stress. It also acts as an immunostimulant by augmenting immunological parameters in the fish skin mucus. Dietary supplementation with SBF essential oil can increase heat stress resistance in fish, which may be associated with higher antioxidant capacity and lower physiological stress in the fish. Based on these results, dietary 100 mg/kg SBF essential oil intake is recommended for rainbow trout rearing. However, further studies are encouraged to assess the efficiency of this dosage under real farm conditions to explore the practical application of SBF essential oil.

## Figures and Tables

**Figure 1 animals-15-02911-f001:**
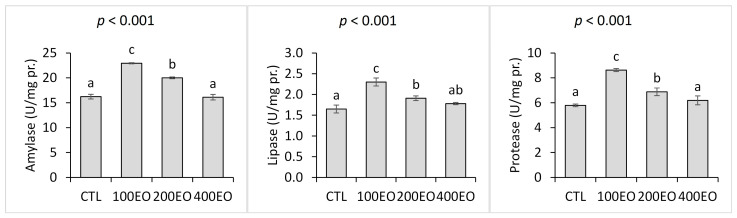
Intestinal digestive enzymes of rainbow trout (mean ± SD) in the CTL, 100EO, 200EO, and 400EO treatments. Different letters above the bars show a significant difference among the treatments (*n* = 3, Duncan test).

**Figure 2 animals-15-02911-f002:**
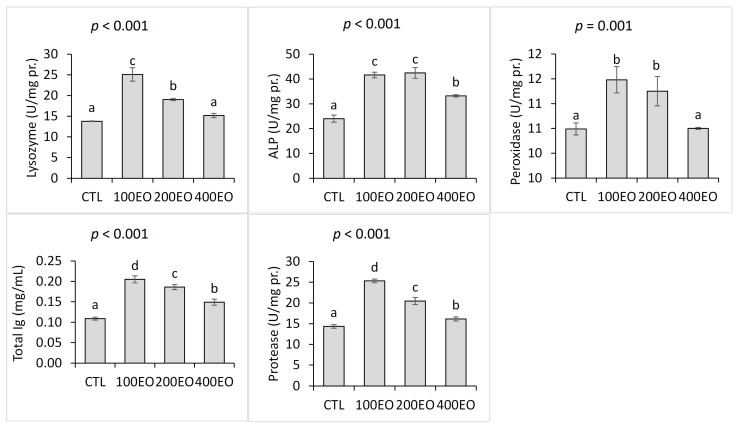
Skin mucus immunological parameters of rainbow trout (mean ± SD) in the CTL, 100EO, 200EO, and 400EO treatments. Different letters above the bars show a significant difference among the treatments (*n* = 3, Duncan test).

**Figure 3 animals-15-02911-f003:**
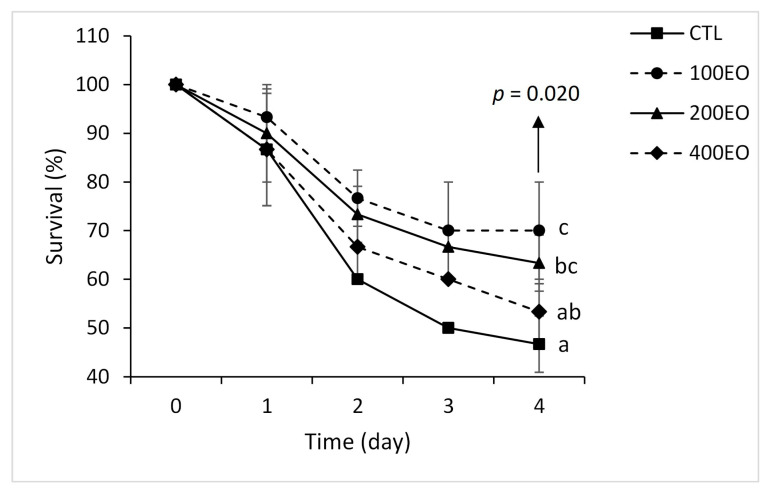
Survival of rainbow trout (mean ± SD) in the CTL, 100EO, 200EO and 400EO treatments following heat stress. Different letters in front of the dots show a significant difference among the treatments at the 4th day post stress (*n* = 3, Duncan test).

**Figure 4 animals-15-02911-f004:**
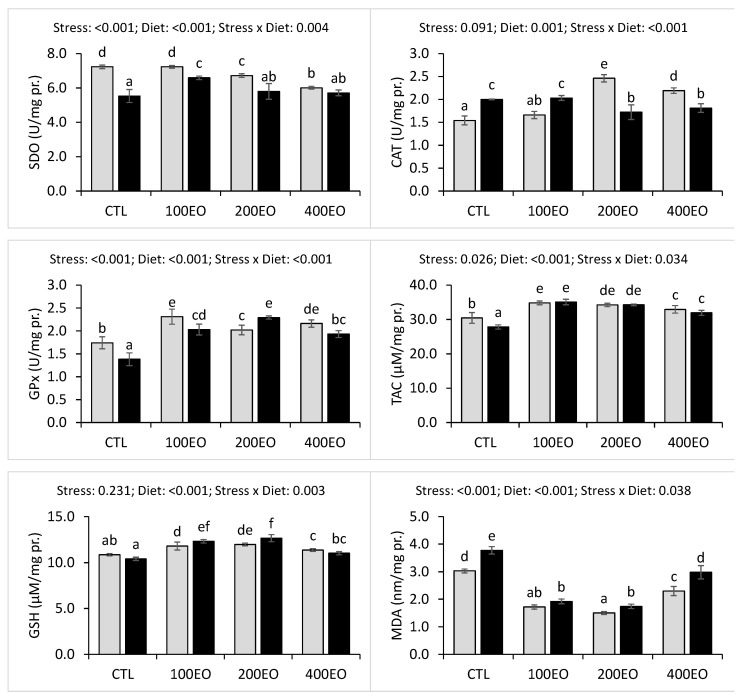
Hepatic antioxidant parameters of rainbow trout (mean ± SD) in the CTL, 100EO, 200EO, and 400EO treatments, before (gray bars) and after (black bars) heat stress. Different letters above the bars show a significant difference among the treatments (*n* =3, Duncan test).

**Table 1 animals-15-02911-t001:** Composition of SBF essential oil determined by GC-MS.

Detected Compounds	Area%	RT (min)
α-Pinene	10.36	5.042
Camphene	2.19	5.327
Δ-3-Carene	2.13	6.624
l-Bornyl acetate	8.85	12.99
trans-Caryophyllene	0.19	15.984
α-Humulene	0.12	16.695
β-Bisabolene	0.07	17.806
14-β-H-Pregna	39.68	27.233
Methyldiethylborane	8.04	29.459
2-Octadecoxyethanol	12.36	30.243
Hexacosane	2.25	34.928
Octacosane	3.06	37.320
Eicosane	10.05	37.331
2-(4-Methylphenyl)indolizine	0.3	44.802
Total	99.67	

**Table 2 animals-15-02911-t002:** Ingredients and chemical composition of the basal diet (g/kg).

Ingredients	g/kg	Proximate Composition	
Fishmeal ^1^	210	Crude protein	425
Poultry slaughterhouse by product	300	Crude lipid	148
Soybean meal	160	Moisture	91.20
Wheat meal	200	Ash	75.8
Sunflower oil	15	Crude fiber	23.5
Soybean oil	20	Gross energy (kcal/g) ^4^	4759
Corn meal	80		
Methionine	3		
Lysine	2		
Vitamin premix ^2^	5		
Mineral premix ^3^	5		

^1^ Pars kilka Co., Mazandaran, Iran (Kilka powder analysis; Protein: 70–72%, Fat: 8–11%, Ash: 11.6%, Moisture: 7–9%). ^2^ Vitamin premix (per kg of diet): vitamin A, 2000 IU; vitamin B_1_ (thiamin), 5 mg; vitamin B_2_ (riboflavin), 5 mg; vitamin B_6_, 5 mg; vitamin B_12_, 0.025 mg; vitamin D_3_, 1200 IU; vitamin E, 63 mg; vitamin K_3_, 2.5 mg; folic acid, 1.3 mg; biotin, 0.05 mg; pantothenic acid calcium, 20 mg; inositol, 60 mg; ascorbic acid (35%), 110 mg; and niacinamide, 25 mg. ^3^ Mineral premix (per kg of diet): MnSO_4_, 10 mg; MgSO_4_, 10 mg; KCl, 95 mg; NaCl, 165 mg; ZnSO_4_, 20 mg; KI, 1 mg; CuSO_4_, 12.5 mg; FeSO_4_, 105 mg; and Co, 1.5 mg. ^4^ Calculated based on 4.3, 5.6, and 9.2 Kcal of energy per gram of carbohydrates, protein, and fat.

**Table 3 animals-15-02911-t003:** Growth performance and survival of rainbow trout (mean ± SD) in the CTL, 100EO, 200EO, and 400EO treatments. Different letters within a row show a significant difference among the treatments (*n* = 3, Duncan test).

	CTL	100EO	200EO	400EO	*p*-Value
Initial weight (g)	4.09 ± 0.11	4.15 ± 0.07	4.10 ± 0.19	4.17 ± 0.05	0.784
Final weight (g)	18.3 ± 0.42 ^a^	23.1 ± 1.21 ^c^	21.5 ± 0.54 ^b^	20.1 ± 0.91 ^b^	0.001
Weight gain (%)	250 ± 21.3 ^a^	459 ± 21.5 ^c^	427 ± 26.5 ^bc^	384 ± 24.2 ^ab^	0.002
Specific growth rate (%/d)	2.50 ± 0.08 ^a^	2.86 ± 0.06 ^c^	2.76 ± 0.08 ^bc^	2.62 ± 0.09 ^ab^	0.003
Feed conversion ratio	1.10 ± 0.03 ^b^	0.89 ± 0.06 ^a^	0.97 ± 0.04 ^a^	1.10 ± 0.08 ^b^	0.004
Survival (%)	100	100	100	100	

**Table 4 animals-15-02911-t004:** Plasma stress indicators of rainbow trout (mean ± SD) in the CTL, 100EO, 200EO, and 400EO treatments, before and after the heat stress. Different letters within a row show a significant difference among the treatments (*n* = 3, Duncan test).

	CTL	100EO	200EO	400EO	Stress	Diet	Stress × Diet
Cortisol (ng/mL)	81.4 ± 5.40	50.7 ± 2.06	68.5 ± 6.20	71.5 ± 3.68	0.001	<0.001	0.916
	98.6 ± 2.76	64.3 ± 10.4	81.6 ± 12.4	89.4 ± 6.93	Before < After	CTL^c^; 100EO ^a^; 200EO ^b^; 400EO ^b^	
Glucose (mg/dL)	51.3 ± 6.81 ^a^	47.0 ± 4.58 ^a^	48.0 ± 7.81 ^a^	45.3 ± 5.51 ^a^	<0.001	0.132	0.008
	135 ± 10.5 ^d^	103 ± 8.00 ^b^	107 ± 13.3 ^bc^	121 ± 15.7 ^cd^			

**Table 5 animals-15-02911-t005:** Plasma and intestinal immunological indicators of rainbow trout (mean ± SD) in the CTL, 100EO, 200EO, and 400EO treatments, before and after the heat stress. Different letters within a row show a significant difference among the treatments (*n* = 3, Duncan test).

		Treatments				ANOVA		
	Stress	CTL	100EO	200EO	400EO	Stress	Diet	Stress × Diet
Plasma								
Lysozyme (U/mL)	Before	33.7 ± 6.61 ^a^	50.1 ± 0.90 ^c^	47.2 ± 1.00 ^bc^	44.2 ± 0.95 ^b^	0.111	0.771	<0.001
	After	30.0 ± 1.82 ^a^	48.1 ± 0.56 ^bc^	44.6 ± 1.06 ^b^	43.9 ± 0.31 ^b^			
ACH50 (U/mL)	Before	140 ± 0.97 ^b^	149 ± 1.80 ^e^	144 ± 1.58 ^cd^	145 ± 0.86 ^d^	<0.001	<0.001	<0.001
	After	124 ± 2.00 ^a^	142 ± 3.79 ^bcd^	143 ± 0.65 ^bcd^	141 ± 1.68 ^bc^			
Total Ig (mg/mL)	Before	20.9 ± 0.28 ^d^	26.6 ± 1.11 ^g^	24.3 ± 0.31 ^f^	23.0 ± 0.23 ^e^	<0.001	<0.001	0.041
	After	15.4 ± 0.60 ^a^	20.3 ± 0.60 ^d^	18.9 ± 0.15 ^c^	17.6 ± 0.14 ^b^			
Intestine								
Lysozyme (U/mg pr.)	Before	8.48 ± 0.19 ^d^	9.95 ± 0.06 ^f^	10.4 ± 0.27 ^g^	9.40 ± 0.16 ^e^	<0.001	<0.001	<0.001
	After	7.15 ± 0.05 ^a^	9.96 ± 0.09 ^f^	8.13 ± 0.22 ^c^	7.61 ± 0.32 ^b^			
Total Ig (mg/g ww.)	Before	19.9 ± 0.64 ^d^	21.5 ± 0.33 ^e^	23.3 ± 0.49 ^f^	20.0 ± 0.61 ^d^	<0.001	<0.001	<0.001
	After	16.7 ± 0.18 ^a^	18.2 ± 0.36 ^c^	17.6 ± 0.09 ^bc^	17.1 ± 0.24 ^ab^			

## Data Availability

The original contributions presented in this study are included in the article. Further inquiries can be directed to the corresponding author.

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
