# Peer review of "Effects of Dietary Supplementation with Abies sibirica Essential Oil on Growth Performance, Digestive Enzymes, Skin Mucus Immunological Parameters, and Response to Heat Stress in Rainbow Trout"

_animals, 2025, doi:10.3390/ani15192911_

Round 1
Reviewer 1 Report
Comments and Suggestions for Authors
The manuscript “Effects of Dietary Supplementation with Essential Oil of Abies sibirica on Growth Performance, Digestive Enzymes, Skin Mucus Immunological Parameters and Response to Heat Stress in Rainbow Trout” is of potential interest to aquaculture nutrition and stress physiology but the manuscript suffers from major conceptual, methodological, and design issues. Here are detailed comments:
These lines "Climate change and earth warming are concerning issues impacting various industries. In the aquaculture industry, these issues are more important in cold-water species, like rainbow trout, Oncorhynchus mykiss." are out of scope of the title as well as objectives of this research.
What is the base for these lines "Herbal feed additives are reliable tools to increase fish growth and health, thereby mitigating the 33 drawbacks of climate changes on fish"?
What is the meaning of having an experiment after 60 days of feeding trial "fish were exposed to a 96-h heat stress and their survival"?
Terminologies such as "100EO" should be abbreviated at first stance when they are used in an article.
How did you extract Abies sibirica essential oil?
The abstract section is poorly written without consideration of statistical analysis. Authors are suggested to rewrite the abstract. The language of the manuscript is unacceptable. There are several instances where the sentence is not giving proper meaning.
Introduction section does not provide enough background of the research. It is very small, and critical links are missing due to which there is no connection between hypothesis and objectives.
Line 64-66, rephrase and rewrite.
What is Scots pine?
Why authors are presenting results "The results indicated that 14-β-H-Pregna (39.68%), 2-Octadecoxyethanol (12.36%), α-pinene (10.36%) and eicosane (1.05%) were identified as the main components of the essential oil" in the Material and method section.
This should be corrected "SBF essential oil was incorporated into the diet at four different levels" as in actual there are only 3 levels of SBF and one is control without SBF.
Why did you use "meat grinder" to pelletize the feed? Authors are instructed to add the specification of this machine in the MS.
Authors are instructed to add the fiber content, gross energy and digestible energy of the diet in the feed proximate composition.
Give the specification of submersible heaters.
There is no proper methodology given for feed preparation, the addition of SBF in feed and other important details which is essential to know the uniform mixing of SBF in the feed.
The design (unclear sample sizes, and insufficient justification for the heat stress protocol) does not provide sufficient robustness for reliable conclusions.
The reported GCMS profile includes compounds inconsistent with the known phytochemistry of Abies sibirica, raising concerns about the reliability of the additive characterization.
These sentences rasises serius concern of not understanding the design clearly and simply giving false statement "plasma and gut biochemical and immunological parameters, along with hepatic antioxidant parameters, were analyzed using repeated measures two-way ANOVA (diet × stress) in conjunction with Duncan's tests. The heat stress was done after the trail so there is no control. Then how come two-way ANOVA (diet × stress) will fit here. Authors need to give clear explanation for selecting 2-way ANOVA here.
What is the base for this "Based on these results, dietary 100 mg/kg SBF essential oil is recommended for rainbow 406 trout rearing"?
Comments on the Quality of English Language
There are numerous grammatical errors, awkward phrasing, and redundancies are present throughout the manuscript which makes it difficult to understand the sentences.
Author Response
The manuscript “Effects of Dietary Supplementation with Essential Oil of Abies sibirica on Growth Performance, Digestive Enzymes, Skin Mucus Immunological Parameters and Response to Heat Stress in Rainbow Trout” is of potential interest to aquaculture nutrition and stress physiology but the manuscript suffers from major conceptual, methodological, and design issues. Here are detailed comments:
These lines "Climate change and earth warming are concerning issues impacting various industries. In the aquaculture industry, these issues are more important in cold-water species, like rainbow trout, Oncorhynchus mykiss." are out of scope of the title as well as objectives of this research.
Response: Thank you for your comment. We chose to challenge the fish with heat stress due to ongoing climate changes and the increasing occurrence of heat waves. Therefore, we believe this statement is necessary to capture the reader’s attention regarding this relevant context.
What is the base for these lines "Herbal feed additives are reliable tools to increase fish growth and health, thereby mitigating the 33 drawbacks of climate changes on fish"?
Response: Thank you. Since we used a herbal feed additive in this study to enhance fish resistance to heat stress, we included this statement. We have now added supporting references in the Introduction to substantiate this claim.
What is the meaning of having an experiment after 60 days of feeding trial "fish were exposed to a 96-h heat stress and their survival"?
Response: Thank you. We first fed the fish diets supplemented with essential oil (EO) for 60 days to evaluate growth and biochemical responses. Subsequently, the fish were subjected to a 96-hour heat stress to assess survival and biochemical responses under stress conditions.
Terminologies such as "100EO" should be abbreviated at first stance when they are used in an article.
Response: Thank you for the suggestion. As stated in line 118-119, these terms were defined as codes rather than abbreviations: SBF essential oil was incorporated into the diet at three levels—100 (100EO), 200 (200EO), and 400 (400EO) mg/kg. We consider these as treatment codes rather than abbreviations.
How did you extract Abies sibirica essential oil?
Response: Thank you. The essential oil used in this study was commercially obtained. The producer’s name is mentioned in the Methods section, line 11.
The abstract section is poorly written without consideration of statistical analysis. Authors are suggested to rewrite the abstract.
Response: Thank you. We have added p-values to the Abstract and improved its overall writing style.
The language of the manuscript is unacceptable. There are several instances where the sentence is not giving proper meaning.
Response: Thank you. we revised the manuscript language and many errors are fixed now.
Introduction section does not provide enough background of the research. It is very small, and critical links are missing due to which there is no connection between hypothesis and objectives.
Response: Thank you. We agree and have revised this section by adding more information on the benefits of essential oils and other herbal additives for enhancing trout growth and heat resistance (line 88-95). Additionally, we have clarified our hypothesis (line 102-106).
Line 64-66, rephrase and rewrite.
Response: Thank you. This section has been revised (line 74-78).
What is Scots pine?
Response: Thank you. This was an error on our part and has been corrected.
Why authors are presenting results "The results indicated that 14-β-H-Pregna (39.68%), 2-Octadecoxyethanol (12.36%), α-pinene (10.36%) and eicosane (1.05%) were identified as the main components of the essential oil" in the Material and method section.
Response: Thank you. This sentence describes the characteristics of the SBF essential oil. We included it in the Methods section, consistent with how we presented characteristics of other materials such as fish meal and diets.
This should be corrected "SBF essential oil was incorporated into the diet at four different levels" as in actual there are only 3 levels of SBF and one is control without SBF.
Response: Thank you. We have revised this as suggested (line 117-119).
Why did you use "meat grinder" to pelletize the feed? Authors are instructed to add the specification of this machine in the MS.
Response: Thank you. This is the equipment available to us for producing feed pellets. Its specifications have been added to line 121-122.
Authors are instructed to add the fiber content, gross energy and digestible energy of the diet in the feed proximate composition.
Response: Thank you. These values have been added to Table 2.
Give the specification of submersible heaters.
Response: Thank you. This information has been added to line 163-164.
There is no proper methodology given for feed preparation, the addition of SBF in feed and other important details which is essential to know the uniform mixing of SBF in the feed.
Response: Thank you. We assure you that the ingredients were thoroughly mixed. Additional details have been added for clarification in lines 127-126.
The design (unclear sample sizes, and insufficient justification for the heat stress protocol) does not provide sufficient robustness for reliable conclusions.
Response: Thank you. The heat stress protocol was based on a previous study on the same species, which is properly cited (line 166). The sample size (n) was 3; we sampled three fish per tank at each sampling time and pooled subsamples to obtain three replicates per treatment. This has been clarified in the revised manuscript (line 176-177 and 193-194).
The reported GCMS profile includes compounds inconsistent with the known phytochemistry of Abies sibirica, raising concerns about the reliability of the additive characterization.
Response: Thank you for this comment. Please note that we used a reference laboratory for GC analysis and the results were double checked. The differences in the EO composition compared to the previous reports can be due to different geographical location of Fir tree, different fir needle collection, different extraction methods and …
These sentences raise serious concern of not understanding the design clearly and simply giving false statement "plasma and gut biochemical and immunological parameters, along with hepatic antioxidant parameters, were analyzed using repeated measures two-way ANOVA (diet × stress) in conjunction with Duncan's tests. The heat stress was done after the trail so there is no control. Then how come two-way ANOVA (diet × stress) will fit here. Authors need to give clear explanation for selecting 2-way ANOVA here.
Response: Thank you. Although there was no “negative control” after stress, two-way ANOVA can still be applied (4 dietary levels × 2 sampling times). An example of this analysis is found in the following paper: https://doi.org/10.1016/j.aquaculture.2019.734668.
What is the base for this "Based on these results, dietary 100 mg/kg SBF essential oil is recommended for rainbow 406 trout rearing"?
Response: Thank you. this treatment had the highest growth, heat resistance and the best physiological parameters among the others. So, we recommended this.
Reviewer 2 Report
Comments and Suggestions for Authors
I have a few questions for the authors:
1) How did the authors determine the heat stress duration of 96 hours?
2) On line 144, 'at each sampling time' – could you please clarify what this term means? How many samples were taken? The previous paragraph mentions only one sample taken after 24 hours of heat stress.
3) Was there an adaptation period for the fish to the new environment and feed?
4) Did the fish show a reduction in feed intake in the first few days of being fed the oil supplement? Was the feeding level unchanged?
5) Did the authors think about checking the tissues and organs of trout that had been exposed to heat stress and had been given feed that contained OE to see what effect these things had on the appearance of cells?
Author Response
I have a few questions for the authors:
1) How did the authors determine the heat stress duration of 96 hours?
Response: Thank you. It was based on a previous study on trout which exposed the fish to 25 C for 96 h. the reference was already added. We revised the text for clarification.
2) On line 144, 'at each sampling time' – could you please clarify what this term means? How many samples were taken? The previous paragraph mentions only one sample taken after 24 hours of heat stress.
Response: Thank you. we meant before and after the heat stress. It was revised in the text.
3) Was there an adaptation period for the fish to the new environment and feed?
Response: Thank you. yes, please check line 141.
4) Did the fish show a reduction in feed intake in the first few days of being fed the oil supplement? Was the feeding level unchanged?
Response: Thank you. No, the fish fed well after the acclimation period, even in the 400EO treatment.
5) Did the authors think about checking the tissues and organs of trout that had been exposed to heat stress and had been given feed that contained OE to see what effect these things had on the appearance of cells?
Response: Thank you. This was out of the scope of the present paper. We focused on biochemical parameter.
Reviewer 3 Report
Comments and Suggestions for Authors
Reviewer Comments:
L80: Could the addition of plant-based feed additives potentially cause feed refusal or reduced palatability in fish? Are there any documented reports in the literature?
L100: After SBF essential oil is incorporated into the feed and processed, do the active compounds remain present? If so, in what amounts?
L138: Did the authors evaluate the physiological parameters of fish after being returned to 17 °C following heat stress? This would help assess the fish’s regulatory ability after experiencing heat stress. If such data are available, I strongly recommend including them. If not, the authors should discuss this aspect in detail in the Discussion section.
Table 3: What could be the possible reasons for the decrease in body weight when the essential oil supplementation level increased?
L238–241: This section should be presented as part of the results discussion, rather than as a note under Table 3.
L329–337: Indeed, short-term supplementation promoted fish growth in the experiment. However, I recommend that the authors clarify that this was under laboratory conditions. In practical aquaculture, factors such as fish size, stocking density, and production cycles (which are usually longer than experimental periods) must be considered. How such feed additives would perform under real farming conditions requires further evaluation. I suggest the authors keep this in mind throughout the Discussion section.
L366: Could stimulating nonspecific immunity through feed additives potentially lead to “immune fatigue”? This is often difficult to assess in short-term trials. I recommend that the authors elaborate on this point.
L406: The authors should discuss why different SBF supplementation levels led to varying responses across parameters. Why was the effect most pronounced at 100 mg/kg SBF instead of increasing with higher supplementation levels?
I also recommend that the authors discuss what the significant interaction between stress and diet represents, as this does not appear to have been addressed in the Discussion.
Author Response
L80: Could the addition of plant-based feed additives potentially cause feed refusal or reduced palatability in fish? Are there any documented reports in the literature?
Response: Thank you. We did not observe any feed refusal in EO-treated fish. To our knowledge, herbal additives at such low levels do not reduce palatability or feed intake.
L100: After SBF essential oil is incorporated into the feed and processed, do the active compounds remain present? If so, in what amounts?
Response: Thank you for this insightful question. We did not measure essential oil content in the feed, as this is uncommon in aquaculture studies. Additionally, no established method exists for extracting essential oil from feed, and interference from other compounds complicates GC analysis.
L138: Did the authors evaluate the physiological parameters of fish after being returned to 17 °C following heat stress? This would help assess the fish’s regulatory ability after experiencing heat stress. If such data are available, I strongly recommend including them. If not, the authors should discuss this aspect in detail in the Discussion section.
Response: Thank you. Unfortunately, we did not monitor fish during recovery. We have added a discussion on this point (line 485-488) as recommended.
Table 3: What could be the possible reasons for the decrease in body weight when the essential oil supplementation level increased?
Response: Thank you. First, EO-treated fish showed higher growth than controls. However, growth decreased at 200EO and 400EO compared to 100EO. These higher doses also showed lower digestive enzyme activities. While the exact mechanism is unclear, it is common for fish to exhibit an optimal inclusion level for feed additives, beyond which performance declines.
L238–241: This section should be presented as part of the results discussion, rather than as a note under Table 3.
Response: Thank you. This is part of the Results section, not a footnote. We have separated it from the table for clarity.
L329–337: Indeed, short-term supplementation promoted fish growth in the experiment. However, I recommend that the authors clarify that this was under laboratory conditions. In practical aquaculture, factors such as fish size, stocking density, and production cycles (which are usually longer than experimental periods) must be considered. How such feed additives would perform under real farming conditions requires further evaluation. I suggest the authors keep this in mind throughout the Discussion section.
Response: Thank you. We have included this statement in the Conclusion section (line 498-500).
L366: Could stimulating nonspecific immunity through feed additives potentially lead to “immune fatigue”? This is often difficult to assess in short-term trials. I recommend that the authors elaborate on this point.
Response: Thank you. We agree and have addressed this point in line 456-458..
L406: The authors should discuss why different SBF supplementation levels led to varying responses across parameters. Why was the effect most pronounced at 100 mg/kg SBF instead of increasing with higher supplementation levels?
Response: Thank you. It is common for fish to respond to graded feed additive levels with an optimal inclusion point, beyond which physiological responses may diminish or change.
I also recommend that the authors discuss what the significant interaction between stress and diet represents, as this does not appear to have been addressed in the Discussion.
Response: Thank you. We have addressed this where biologically relevant. Sometimes significant interactions have statistical rather than biological interpretations. For example, glucose showed a significant stress × diet interaction because stress altered the diet’s effect on glucose: before stress, diet had no effect, but after stress, some SBF treatments reduced glucose compared to control, indicating mitigation of stress-induced glucose elevation. In the discussion, we wrote this fact “dietary SBF mitigated stress and energy expenditure after the stress”, although we did not directly emphasize on “interaction”. Similarly, plasma lysozyme activity showed interaction due to changes in differences between treatments before and after stress. We have discussed these points in the manuscript, emphasizing that dietary SBF improved immune parameters both before and after stress.
Round 2
Reviewer 1 Report
Comments and Suggestions for Authors
The manuscript has been substantially revised, and authors have addressed all the queries satisfactorily. I have no further query.
Reviewer 3 Report
Comments and Suggestions for Authors
The authors have adequately addressed my comments, and the necessary additions have been incorporated into the revised manuscript. The manuscript, after revision, is now suitable for publication.